# Genome-Based Characterization of Hybrid Shiga Toxin-Producing and Enterotoxigenic *Escherichia coli* (STEC/ETEC) Strains Isolated in South Korea, 2016–2020

**DOI:** 10.3390/microorganisms11051285

**Published:** 2023-05-15

**Authors:** Woojung Lee, Min-Hee Kim, Soohyun Sung, Eiseul Kim, Eun Sook An, Seung Hwan Kim, Soon Han Kim, Hae-Yeong Kim

**Affiliations:** 1Division of Food Microbiology, National Institute of Food and Drug Safety Evaluation, Ministry of Food and Drug Safety, Cheongju 28159, Republic of Korea; woojungluv79@korea.kr (W.L.); minhee960303@korea.kr (M.-H.K.);; 2Institute of Life Sciences & Resources, Department of Food Science and Biotechnology, Kyung Hee University, Yongin 17104, Republic of Korea

**Keywords:** Shiga toxin-producing *E. coli* (STEC), enterotoxigenic *E. coli* (ETEC), whole-genome sequencing, comparative genomics, virulence gene, plasmid, bacteriophages

## Abstract

The global emergence of hybrid diarrheagenic *E. coli* strains incorporating genetic markers from different pathotypes is a public health concern. Hybrids of Shiga toxin-producing and enterotoxigenic *E. coli* (STEC/ETEC) are associated with diarrhea and hemolytic uremic syndrome (HUS) in humans. In this study, we identified and characterized STEC/ETEC hybrid strains isolated from livestock feces (cattle and pigs) and animal food sources (beef, pork, and meat patties) in South Korea between 2016 and 2020. The strains were positive for genes from STEC and ETEC, such as *stx* (encodes Shiga toxins, Stxs) and *est* (encodes heat-stable enterotoxins, ST), respectively. The strains belong to diverse serogroups (O100, O168, O8, O155, O2, O141, O148, and O174) and sequence types (ST446, ST1021, ST21, ST74, ST785, ST670, ST1780, ST1782, ST10, and ST726). Genome-wide phylogenetic analysis revealed that these hybrids were closely related to certain ETEC and STEC strains, implying the potential acquisition of Stx-phage and/or ETEC virulence genes during the emergence of STEC/ETEC hybrids. Particularly, STEC/ETEC strains isolated from livestock feces and animal source foods mostly exhibited close relatedness with ETEC strains. These findings allow further exploration of the pathogenicity and virulence of STEC/ETEC hybrid strains and may serve as a data source for future comparative studies in evolutionary biology.

## 1. Introduction

*Escherichia coli* is commonly regarded as a nonpathogenic beneficial inhabitant of the gastrointestinal tract. However, several pathogenic strains have acquired specific virulence factors that are responsible for various intestinal and extraintestinal diseases, including diarrhea, acute inflammation, hemorrhagic colitis, urinary tract infections, septicemia, and neonatal meningitis. Diarrheagenic *Escherichia coli* (DEC) causes 30–40% of acute diarrhea episodes in children <5 years in developing countries [1]. According to the WHO Global Burden of Foodborne Diseases report, >300 million illnesses and nearly 200,000 deaths are caused by DEC globally each year [2]. Major diarrheagenic *E. coli* (DEC) strains are subdivided into several pathotypes based on the presence of specific virulence traits directly related to disease development [3,4,5,6]. The DEC pathotypes include enteropathogenic *E. coli* (EPEC), Shiga toxin-producing *E. coli* (STEC), enterotoxigenic *E. coli* (ETEC), enteroinvasive *E. coli* (EIEC), and enteroaggregative *E. coli* (EAEC). Many of these pathotypes are foodborne pathogens that raise public health concerns and cause several outbreaks in industrialized and developing countries [7,8,9].

STEC and ETEC are major causes of diarrhea in humans and animals worldwide. STEC is characterized by the presence of the Shiga toxin 1 or 2 genes (*stx*_1_ or *stx*_2_), which are generally acquired by a lambda-like bacteriophage [10]. Shiga toxins 1 and 2 (Stx1 and Stx2, respectively) differ in their virulence and host specificity, with Stx2 being most commonly associated with severe illnesses (hemolytic uremic syndrome (HUS), hospitalization, and bloody diarrhea) in humans [11,12]. ETEC is characterized by its ability to produce either a heat-labile (LT) or heat-stable (ST) enterotoxin and carries a diverse set of colonization factors (CFs) for adherence to the intestinal epithelium [13]. It is a major cause of diarrhea among children living in and tourists traveling to developing countries.

Hybrid DEC strains that combine genetic markers belonging to different pathotypes have emerged worldwide and are a public health concern [14]. Numerous virulence markers are frequently carried on mobile genetic elements (MGEs), such as phages and plasmids, allowing the transmission of virulence genes via horizontal gene transfer, leading to the emergence of hybrid pathotypes [3,15,16,17,18]. Hybrid *E. coli* strains comprising genetic markers of different pathotypes have been identified owing to the technological advances that provide a better understanding of the genomic and virulence mechanisms of DEC [19].

The most well-documented example is the *E. coli* O104:H4 strain, which caused a severe outbreak of acute gastroenteritis and HUS in Germany in 2011 [20]. This strain produced Stx2, a signature feature of the STEC pathotype, and it carried a plasmid containing the genes encoding aggregative adherence fimbriae (AAF), which mediate aggregative adherence in EAEC [21,22,23]. Furthermore, hybrids of STEC and ETEC strains (STEC/ETEC) have been recently reported in various countries, including Bangladesh, Sweden, and South Korea, some of which have been associated with diarrheal diseases and HUS in humans [24,25,26,27,28].

Few studies have reported the virulence and antibiotic resistance profiles of STEC/ETEC hybrid strains isolated from livestock feces (cattle and pigs) and animal source foods (beef, pork, and meat patties) in South Korea. This study investigated the genomes of STEC/ETEC hybrid strains to identify the virulence and antibiotic resistance genes they harbored and to determine their phylogenetic position among other *E. coli* strains. The genomic properties of these strains were investigated via real-time PCR and whole-genome sequencing (WGS). Phylogenetic analysis was performed to assess their phylogeny in a collection of *E. coli* strains from diverse pathotypes. Based on our findings, we addressed the potential importance of these hybrid *E. coli* strains for public health.

## 2. Materials and Methods

### 2.1. Bacterial Strains and Serotyping

Pathogenic *E. coli* strains that originated from the Korean Culture Collection for Foodborne Pathogens (Ministry of Food and Drug Safety) were identified. All 1025 pathogenic *E. coli* strains isolated in South Korea between 2016 and 2020 were analyzed. Twenty-seven hybrid Shigatoxigenic and enterotoxigenic *Escherichia coli* (STEC/ETEC) strains were isolated from livestock feces (cattle and pigs) and animal source foods (beef, pork, and meat patties). The strains selected for this study are listed in Table 1. Typical *E. coli* colonies (blue-green color) on 5-bromo-4-chloro-3-indolyl-β-D-glucuronide (BCIG) agar (Oxoid, UK) were sub-cultured on Tryptic Soy Agar (Oxoid, UK) and then incubated at 37 °C for 18–24 h. The isolates were identified using VITEK MS (BioMerieux Inc., Marcy-l’Etoile, France). The serotype was determined by the agglutination of the bacteria with specific somatic (O1 to O181) antisera [Laboratorio de Referencia de *E. coli* (LREC), Lugo, Spain] to identify variants of the somatic (O) antigens [29,30,31].

### 2.2. Antimicrobial Susceptibility Tests

Antimicrobial susceptibility tests were performed using Sensititre KRN6F panels (Trek Diagnostic Systems, Cleveland, OH, USA) following the manufacturer’s instructions. The antimicrobial susceptibility of the isolated strains was determined using the 16 antimicrobials described as follows: amoxicillin–clavulanic acid, ampicillin, cefoxitin, cefotaxime, ceftazidime, cefepime, chloramphenicol, ciprofloxacin, colistin, gentamicin, meropenem, nalidixic acid, streptomycin, sulfisoxazole, tetracycline, and trimethoprim-sulfamethoxazole. The MIC (minimum inhibitory concentration) value of these antimicrobials was determined with the microbroth dilution method. The Clinical and Laboratory Standards Institute guidelines and the U.S. National Antimicrobial Resistance Monitoring System were used to interpret susceptibility results expressed as MICs. For these agents, the degree of increase in resistance was determined by referring to the resistance level of the standard strain, ATCC 25922.

### 2.3. Real-Time PCR-Based Identification of Hybrid Strains

DNA was extracted from the bacterial cultures using automated equipment (EZ1 Advanced XL, Qiagen, Germantown, MD, USA) according to the manufacturer’s instructions. The extracted DNA was used as a template for real-time PCR, which was performed using a PowerCheck^TM^ 20/15 Pathogen Multiplex Real-time PCR kit (Kogene Biotech Co., Ltd., Seoul, Korea) to detect virulence genes. Amplification was performed using an ABI 7500 Fast Real-time PCR system (Applied Biosystems, Waltham, MA, USA) at 50 °C for 2 min for 1 cycle, 95 °C for 10 min for 1 cycle, followed by 40 cycles at 95 °C for 15 s, 60 °C for 1 min. The following genes from different DEC pathogens were detected: *VT1* and *VT2* (STEC); *bfpA* and *eaeA* (EPEC); *LT*, *STh,* and *STp* (ETEC); *aggR* (EAEC); *ipaH* (EIEC).

### 2.4. Genome Sequencing, Assembly, and Annotation

Genomic DNA was extracted using the MagListo^TM^ 5M Genomic DNA Extraction Kit (Bioneer, Daejeon, Korea) according to the manufacturer’s protocol. DNA integrity and concentration were determined using standard agarose gel electrophoresis and a Qubit^TM^ 3.0 Fluorometer (Life Technologies, Carlsbad, CA, USA), respectively. A DNA library was prepared using a Nextera DNA Flex Library Prep Kit (Illumina, San Diego, CA, USA). Sequencing was performed using a MiSeq sequencing system (Illumina) and MiSeq Reagent Kit v3 (600-cycle) (Illumina). The contigs (FASTQ sequence files) were assembled de novo using the CLC Workbench (version 12.0; Qiagen, Hilden, Germany). To obtain high-quality data and determine the complete genomic sequence, the hybrid genome was assembled using additional long-read sequence data obtained from PacBio Sequel (Pacific Bioscience, Menlo Park, CA, USA). Hybrid assembly of raw FASTQ PacBio sequence long-read sequence data and Illumina MiSeq short-read FASTQ sequence data was performed using Unicycler (v0.4.9; https://github.com/rrwick/Unicycler (accessed on 1 January 2023); default settings). The assembled genome was annotated using the Rapid Annotation using Subsystem Technology (RAST) toolkit in the PATRIC genome annotation web service (v3.6.12).

### 2.5. DNA Sequence and Bioinformatics Analysis

Virulence factors and antimicrobial resistance genes were identified using the virulence factor database and ResFinder v4.1 [32,33], respectively. Adhesion and colonization factors in the human intestine required for STEC pathogenesis, such as the locus of adhesion and autoaggregation (LAA)-related genes, were identified using the Basic Local Alignment Search Tool (BLAST). Mobile genetic elements were identified using MobileElementFinder (https://cge.cbs.dtu.dk/services/MobileElementFinder/) (accessed on 22 February 2023) [34]. CRISPRCasFinder (https://crisprcas.i2bc.paris-saclay.fr/CrisprCasFinder/Index) (accessed on 23 April 2023) was used to analyze the CRISPRs and *cas* genes [35]. Whole-genome multilocus sequence typing (wgMLST) was performed with BioNumerics (v8.0; Applied Maths, Sint-Martens-Latem, Belgium), where 17,380 loci from *E. coli* and *Shigella* were included for analysis. In addition, the web-based serotyping tool, SerotypeFinder 2.0 [36], was used to predict the antigen profiles of *E. coli* strains.

### 2.6. Prophage Prediction and Analysis

Bacteriophage sequences within the unique genome sequence were identified using PHAge Search Tool Enhanced Release (PHASTER) [37]. PHASTER was used to predict putative prophage regions as “intact (score > 90),” “questionable (score 70–90),” or “incomplete (score < 70)” based on the proportion of phage-related genes in the identified phage region of the assembled hybrid genome. The extracted prophage sequences were annotated to identify virulence genes using the RAST toolkit in the PATRIC genome annotation web service (v3.6.12).

### 2.7. Identification of Plasmid-Associated Sequences

Plasmid features of the assembled hybrid genomes were analyzed using PlasmidFinder 2.1 [38]. The threshold and minimum coverage for identification were set to 95% and 60%, respectively. The identification was based on the detection of replicon sequences belonging to several known plasmid incompatibility (Inc) groups. The extracted plasmid sequences were annotated to identify virulence genes using the RAST toolkit in the PATRIC genome annotation web service (v3.6.12).

### 2.8. Phylogenetic Analysis and Population Structure Analysis

Comparative genomic analysis was performed on 27 STEC/ETEC hybrid strains isolated from livestock feces (cattle, pigs) and animal source foods (beef, pork, meat patties) in South Korea and 187 pathogenic *E. coli* strains. The genomic sequences of 160 strains isolated from food and the environment in South Korea and 27 other pathogenic *E. coli* strains are available at the National Center for Biotechnology Information (NCBI). The genomes analyzed in this study are summarized in Appendix A. The pan-genome was analyzed using the bacterial pan-genome analysis (BPGA) tool (v1.3; default parameters). The USEARCH tool was used for clustering with 95% sequence identity as the cut-off value. The phylogenetic tree was clustered using the neighbor-joining method and visualized using the Interactive Tree of Life (iTOL) v6. The population structure analysis was performed using RhierBAPs [39].

## 3. Results

### 3.1. Genome Assemblies of STEC/ETEC Hybrids

Twenty-seven STEC/ETEC hybrid strains isolated from livestock feces and animal source foods in South Korea were sequenced. All strains had one chromosome and one plasmid. The genomic characteristics of the hybrid STEC/ETEC strains are summarized in Table 1. The genome lengths of these isolates ranged from 5,064,469 to 5,865,149 bp, with coverage ranging from 126× to 524×. In addition, the G + C content of the genomes of these strains was between 50.3% and 50.9%, the length of the coding DNA sequences (CDSs) was between 5081 and 6141 bp, and the number of tRNA and rRNA genes was 82–107 and 22, respectively.

### 3.2. In Silico Identification of Virulence and CRISPR-Associated (Cas) Genes

The initial screening of pathogenic *E. coli* strains was conducted using real-time PCR. The hybrid STEC/ETEC isolates harbored both Shiga toxin 2 (*stx2*) and heat-stable enterotoxins (*est*) encoding genes. Subsequently, we performed virulence gene mapping to identify the various virulence factors present in the STEC/ETEC hybrid genomes (Figure 1). Multiple virulence factors have been implicated in *E. coli* pathogenesis, including the Shiga toxin and enterotoxins, as well as other factors such as adhesion factors, colonization factors (CFs), non-LEE-encoded TTSS effectors, and secretion systems. Detailed results of virulence gene mapping of these hybrid genomes are shown in Appendix A. Importantly, we also detected genes encoding LAA, such as *hes*, *iha*, *lesP*, and *agn43*, which are related to STEC pathogenicity. The most prevalent gene was *iha* (59.3%), followed by *agn43*, *hes*, and *lesP* that were present in 25.9, 18.5, and 11.1% of the hybrid strains, respectively. In addition, the CRISPRFinder server identified a type I CRISPR/Cas system in all hybrid strains. Additionally, most of the STEC/ETEC hybrid strains (26/27) identified the type I-E system. All hybrid strains harbored the *cas3* gene, which is the signature of type I CRISPR/Cas systems, responsible for target DNA cleavage and degradation [40]. Furthermore, most of the STEC/ETEC hybrid strains (26/27) harbored the *cas1*, *cas2*, *cas5*, *cas6*, and *cas7* genes as well as the *cas3* gene.

### 3.3. In Silico Identification of Antimicrobial Resistance Genes

ResFinder was used to predict antimicrobial resistance genes in the hybrid genomes (Figure 2). Most of the genomes (16/27) contained at least two antibiotic (ampicillin, piperacillin, streptomycin, and ticarcillin) resistance genes, whereas nine isolates were negative for them. The comprehensive results of the antimicrobial resistance gene mapping of the hybrid *E. coli* genomes are shown in Appendix A. High rates of resistance gene were observed for tetracycline (55.6%), doxycycline (55.6%), chloramphenicol (51.9%), sulfamethoxazole (51.9%), florfenicol (48.1%), streptomycin (48.1%), amoxicillin (44.4%), ampicillin (44.4%), and piperacillin (44.4%). Especially, the *tetA* and *tetB* genes were found at the highest frequency in hybrid *E. coli* strains. We additionally performed antimicrobial susceptibility tests to determine the phenotypic profile of antimicrobial resistance in STEC/ETEC hybrid strains. The phenotypic profile of antimicrobial resistance is described in detail in Appendix A. Comparing the WGS-based AMR genotype to the antimicrobial susceptibility testing-based phenotype for 13 antibiotics (ampicillin, cefepime, cefoxitin, cefotaxime, ceftazidime, chloramphenicol, ciprofloxacin, colistin, nalidixic acid, meropenem, gentamicin, streptomycin, tetracycline) revealed concordant results for 22 of the 27 STEC/ETEC hybrid strains (81.5%). In five STEC/ETEC hybrid strains, only antibiotic resistance genes were identified, but no phenotypes. Although WGS can provide more information about isolates, genomic approaches cannot always predict phenotypes because the level of gene expression and protein production from identified genes may differ between strains. Bacteria have gene-silencing mechanisms, and mutations may generate stop codons in the data [41].

### 3.4. Serotyping and Sequence Types of the Hybrids

The serotype and sequence type results for the 27 hybrid STEC/ETEC isolates are summarized in Table 2. Based on in silico serotyping, the 27 hybrid *E. coli* strains belonged to eight distinct O:H serogroups [O100:H30 (*n* = 7), O8:H9 (*n* = 5), O168:H8 (*n* = 5), O155:H21 (*n* = 4), O2:H25 (*n* = 3), O141:H29 (*n* = 1), O148:H7 (*n* = 1), and O174:H2 (*n* = 1)]. Especially, the O100:H30 serogroup was found at the highest frequency in hybrid *E. coli* strains. The hybrid strains represented diverse sequence types [ST446 (*n* = 7), ST1021 (*n* = 5), ST21 (*n* = 4), ST74 (*n* = 4), ST785 (*n* = 1), ST670 (*n* = 2), ST1780 (*n* = 1), ST1782 (*n* = 1), ST10 (*n* = 1), and ST726 (*n* = 1)].

### 3.5. Phage Characterization

To investigate the phage-mediated horizontal gene transfer of *stx* genes in hybrid STEC/ETEC isolates, we identified the bacteriophage sequences using PHASTER. The results obtained for the 27 hybrid STEC/ETEC strains are summarized in Table 3. The presence of the majority of *stx2* gene sequences was confirmed from phage sequence regions corresponding to “intact” (26/27). Additionally, it was confirmed that the sequences corresponding to the *stx2* gene in one STEC/ETEC hybrid strain genome (MFDS1012367; score 90) were found in the “questionable” phage region.

### 3.6. Plasmid-Associated Sequence

To investigate the plasmid-mediated horizontal gene transfer of *est* in the hybrid STEC/ETEC isolates, we analyzed plasmid-associated sequences using PlasmidFinder 2.1. The plasmid replication results for the 27 hybrid STEC/ETEC isolates are summarized in Table 3. PlasmidFinder identified several plasmid replicon sequences of known Inc groups in all the STEC/ETEC genomes. We elucidated that each of the 27 genomes had an IncFIl or IncFIB plasmid origin and in some cases, such as MFDS1016200, both. The genes encoding heat-stable enterotoxin STa (*estA*) or STb (*estB*) were placed in the same contig as IncFIl and IncFIB. The mobile genetic elements proximal to *estA* and *estB* are shown in Appendix A.

### 3.7. Phylogenetic Analysis and Population Structure Analysis

The genomes of 187 isolates, comprising 41 STEC, 46 ETEC, 72 EPEC, 18 EAEC, and 10 EIEC strains, were used for the phylogenetic analysis to determine the genomic relationship between the STEC/ETEC hybrids and other pathogenic *E. coli* isolates. The 187 genome datasets included the sequencing results of 160 pathogenic *E. coli* as well as 27 hybrid STEC/ETEC genomes, which were deposited in the NCBI database. Phylogenetic tree analysis revealed that these hybrids were closely related to certain ETEC (21 strains, 77.8%) and STEC (six strains, 22.2%) strains, implying the potential acquisition of Stx-phages and/or ETEC virulence genes during their emergence (Figure 3A). In addition, the population structure of the 187 genome datasets was defined using the RhierBAPS, which divided the genome datasets into six primary sequence clusters (Bayesian analysis of population structure [BAPS] hierarchical level 1). These were further subdivided into 28 lineages (BAPS level 2) (Figure 3B). The results showed that of the total, 21 hybrid strains closely related to ETEC were divided into six groups (three level 1, five level 2), while the remaining six hybrids correlated with STEC were divided into two groups (two level 1, two level 2).

## 4. Discussion

STEC/ETEC hybrids have been recovered from various sources, including humans, animals, food, and water, some of which have been associated with diarrheal diseases and HUS in humans. In South Korea, the STEC/ETEC hybrid strain was first isolated from a patient suffering from diarrhea in 2014 [23]. Here, we report 27 STEC/ETEC hybrid strains among 1025 pathogenic *E. coli* strains identified in South Korea between 2016 and 2020. This study characterized the virulence and antibiotic resistance genes harbored by these hybrid strains, to further determine their phylogeny among other pathogenic *E. coli* strains. The molecular properties of these strains were investigated using real-time PCR followed by whole-genome sequencing (WGS). Phylogenetic analysis was performed to assess the phylogenetic positions of these hybrids in a diverse collection of pathogenic *E. coli* representing all the major pathotypes.

For the initial molecular characterization of all pathogenic *E. coli* strains, real-time PCR and serotyping were employed. Subsequent WGS analysis of these hybrids yielded results that were consistent with serotyping and the presence of virulence factors. The presence of genes encoding Shiga toxin 2 and heat-stable enterotoxin, namely, *stx2* and *est*, respectively, was confirmed in all 27 STEC/ETEC hybrid strains. Most of the STEC/ETEC hybrid strains among human and animal isolates in Finland [25] harbored the *stx2* gene without the *stx1* gene. In addition, STEC/ETEC hybrid strains from diarrheal patients in South Korea [27] and Sweden [28] harbored the *stx2* gene. However, the majority of STEC/ETEC hybrid strains in livestock of Bangladesh [26] carried the *stx1* gene. Our results suggest that the STEC/ETEC hybrid strains isolated in South Korea that contained the *stx2* gene may be more dangerous to humans. Shiga toxins are major factors contributing to the virulence of STEC; however, adhesion and colonization to the human intestine are required for STEC pathogenesis [42,43,44]. Some STEC strains carry the locus of enterocyte effacement (LEE-positive) [45,46,47], whereas those that do not carry the LEE (LEE-negative) and mainly harbor the locus of adhesion and autoaggregation (LAA) have also been associated with illness [48,49,50,51]. LAA is found either as a “complete” structure with four modules (module I (*hes* and other genes), module II (*iha*, *lesP*, and others genes), module III (*pagC*, *tpsA*, and other genes), and module IV (*agn43* and other genes)) or as an “incomplete” structure if one of the modules is missing [48]. In this study, 27 STEC/ETEC hybrid strains carried one copy of the *stx2* gene, lacked *eae* (*E. coli* attaching and effacing) gene and were LEE-negative STEC strains. Consistent with findings of previous studies, we observed some genes encoding LAA in some of the identified STEC/ETEC hybrid strains, except for seven strains (MFDS 1009773, 1012367, 1014122, 1016183, 1016224, 1016228, and 1016229).

In addition, the colonization of ETEC on the surface of the intestinal epithelium is a critical step in exerting its toxicity [52]. In addition to heat-labile (LT) and/or heat-stable (ST) enterotoxins, colonization factors (CFs) are major virulence factors in ETEC. Once ETEC colonizes the small intestinal epithelia through CFs, effective enterotoxin delivery commences, which is responsible for the secretion of water and electrolytes from the intestinal lumen [53,54]. These factors are referred to as colonization factor antigen I (CFA/I) or coli surface (CS) antigen [55]. The CFA/I of ETEC-related genes, such as *cfa A*, *cfa B*, *cfa C*, *cfa D*, and *cfa E*, was detected in 16 STEC/ETEC hybrid strains.

In addition, the hybrids represented diverse serotypes (O2:H25 (2), O2:H27 (1), O8:H8 (1), O8:H9 (4), O100:H30 (7), O141:H29 (1), O148:H7 (1), O155:H21 (4), O168:H8 (5), and O174:H2 (1)] and sequence types [ST446 (7), ST1021 (5), ST21 (4), ST74 (4), ST785 (1), ST670 (2), ST1780 (1), ST1782 (1), ST10 (1), and ST726 (1)). Previous studies have revealed the diversity of sequence types and serotypes (>40) among STEC/ETEC hybrid strains. This diversity suggests that both the ETEC virulence gene-carrying plasmids and Shiga toxin-containing bacteriophages could spread to a broad range of genetic backgrounds, including serotypes related to more pathogenic disease-causing strains such as O2:H27, O15:H16, O101:H-, O128:H8, and O141:H8 [25].

This study describes the virulence gene transfer of STEC/ETEC hybrid strains isolated from livestock feces and animal source foods in South Korea. It emphasizes that WGS is a powerful tool to analyze bacterial genomes for the presence of regions of MGEs, such as phages and plasmids, in them. Furthermore, the genomic information obtained in this study can significantly contribute to a better understanding of the genomic characteristics of hybrid *E. coli* strains in the future. In further studies, it may be necessary to investigate the genomic and transcriptome characteristics of STEC/ETEC hybrid strains isolated from diverse ecological and geographical sources in Korea.

## 5. Conclusions

In conclusion, we are the first to report the virulence and antibiotic resistance profiles of STEC/ETEC hybrid strains isolated from livestock feces (cattle and pigs) and animal source foods (beef, pork, and meat patties) in South Korea. Through genome-based characterization, we confirmed that virulence markers present in STEC/ETEC pathotypes were carried by MGEs, such as phages and plasmids. In addition, we identified adhesion and colonization factors in the human intestine required for STEC pathogenesis. Most DEC is subdivided into several pathotypes based on the presence of specific virulence traits directly related to disease development [3,4]. Importantly, our results emphasize that the hybrid strains of *E. coli* with STEC and other DEC-associated virulence factors may be more dangerous than STEC alone [14,20,21,22,23,24,25,26,27,28]. Thus, the emergence of hybrid DEC strains may have severe consequences for public health and should be considered in patient care and epidemiological surveillance.

## Figures and Tables

**Figure 1 microorganisms-11-01285-f001:**
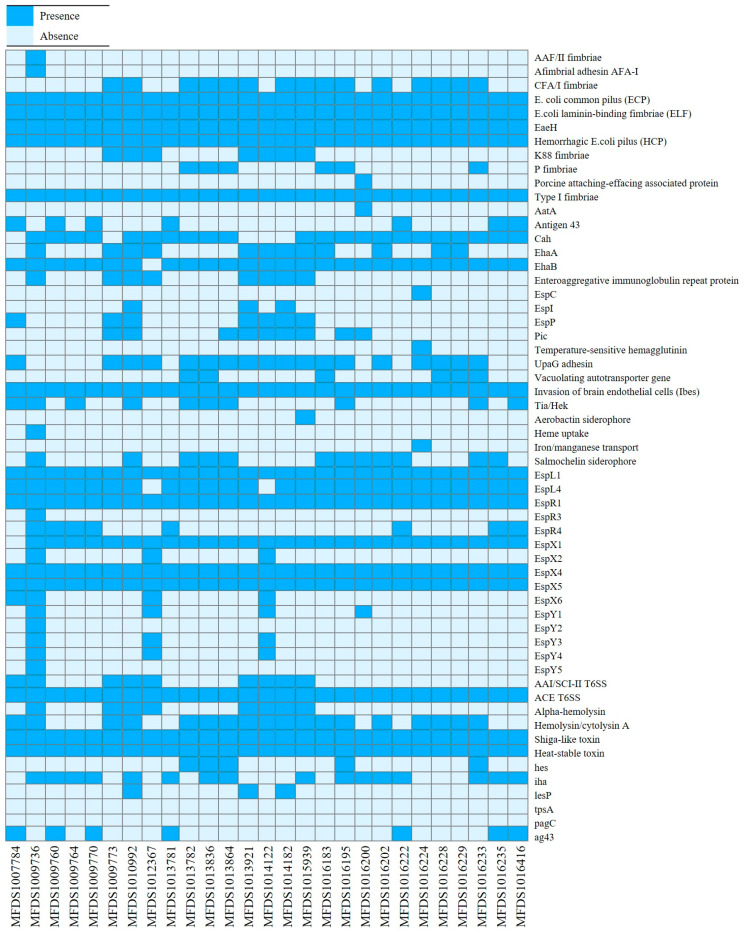
The heatmap of virulence factors across the hybrid Shiga toxin-producing *Escherichia coli* (STEC)/enterotoxigenic *E. coli* (ETEC) strains. Heatmap showing the presence/absence of virulence factors (*y*-axis) among the hybrid STEC/ETEC isolates identified in this study (*x*-axis). The presence of virulence genes is shown in dark blue, whereas the absence is shown in light blue, as indicated in the color key. Heatmap was generated using the gplot (v3.1.3) package in R software (v4.1.3).

**Figure 2 microorganisms-11-01285-f002:**
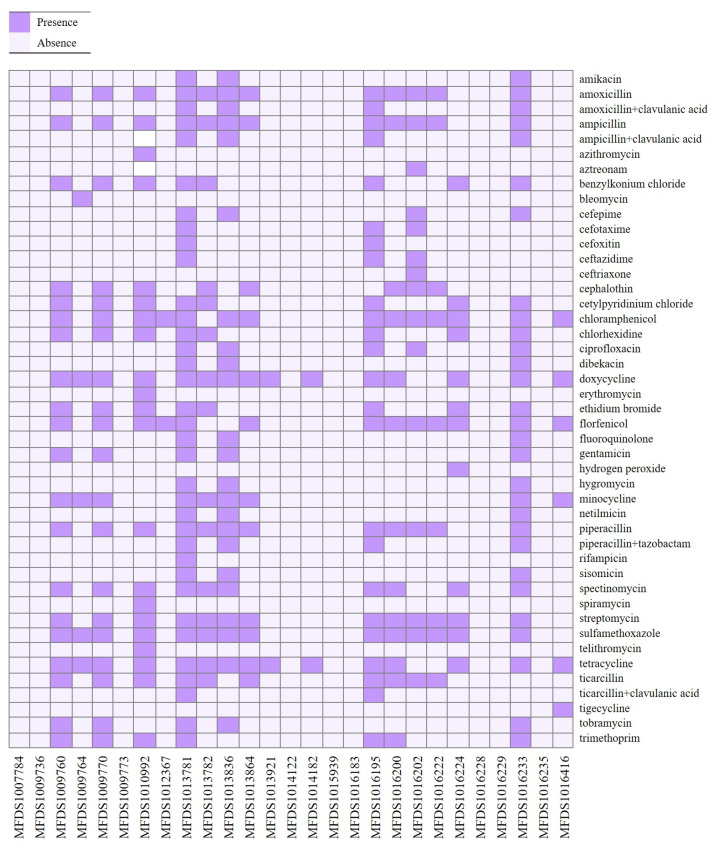
The heatmap of antimicrobial profiles across the hybrid STEC/ETEC strains. Heatmap showing the presence/absence of antimicrobial resistance (*y*-axis) among the hybrid STEC/ETEC isolates identified in this study (*x*-axis). The presence of antimicrobial genes is shown in bright purple, whereas absence is shown in light purple, as indicated in the color key. Heatmap was generated using the gplot (v3.1.3) package in R software (v4.1.3).

**Figure 3 microorganisms-11-01285-f003:**
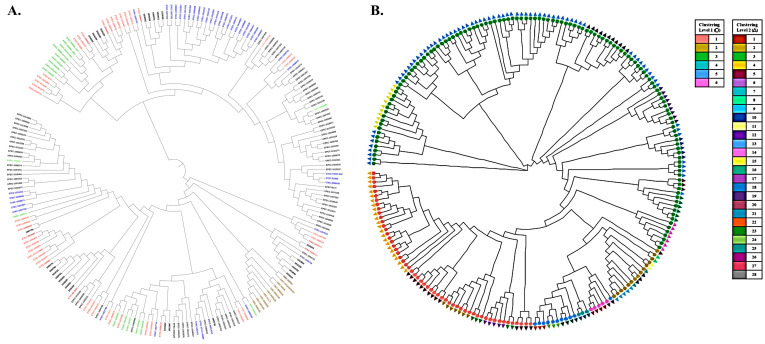
Phylogenetic analysis and population structure analysis of hybrid STEC/ETEC strains. (**A**) Strains are colored based on the pathogenic *E. coli* groups: blue, STEC strains; pink, ETEC strains; gray, EPEC strains; green, EAEC strain; and brown, EIEC strains. The pink and blue bars represent 21 and 6 STEC/ETEC hybrid strains that are closely related to specific ETEC and STEC strains, respectively. (**B**) Sequence clusters (1 to 6) are indicated in the outer colored dot, which are further divided into 28 lineages (inner ring). Of the total, 21 hybrid strains closely related to ETEC were divided into six groups (three level 1, five level 2), while the remaining 6 hybrids correlated with STEC were divided into two groups (two level 1, two level 2).

**Table 1 microorganisms-11-01285-t001:** Summarized characteristics of hybrid Shiga toxin-producing *Escherichia coli* (STEC)/enterotoxigenic *E. coli* (ETEC) strains.

Strain Name	Collection Date	Geographic Location	Isolation Source	Coverage	Contigs	Size (bp)	GC (%)	CDSs	rRNA	tRNA	Accession No.
MFDS1007784	8 June 2016	Jeollanam-do	animal source foods (meat patties)	178	5	5,327,823	50.6	5465	22	107	JAQJCX000000000
MFDS1009736	31 August 2017	Incheon	livestock feces	243	5	5,431,463	50.4	5439	22	93	JAQMUJ000000000
MFDS1009760	20 May 2017	Incheon	livestock feces (pig)	288	8	5,332,414	50.6	5411	22	91	JAQMUK000000000
MFDS1009764	4 July 2017	Incheon	livestock feces (pig)	126	14	5,355,898	50.5	5467	22	89	JAQMUL000000000
MFDS1009770	18 July 2017	Incheon	livestock feces (pig)	182	4	5,322,048	50.6	5419	22	91	JAQMUM000000000
MFDS1009773	25 July 2017	Incheon	livestock feces (cattle)	176	7	5,310,620	50.6	5419	22	100	JAQMUO000000000
MFDS1010992	31 January 2018	Jeollabuk-do	animal source foods (beef)	251	6	5,808,504	50.4	5998	22	100	JAQMUP000000000
MFDS1012367	28 December 2018	Jeju-do	animal source foods (beef)	177	5	5,253,992	50.7	5518	22	94	JAQMUR000000000
MFDS1013781	12 February 2019	Chungcheongnam-do	livestock feces (pig)	278	7	5,392,379	50.6	5517	22	89	JAQMUS000000000
MFDS1013782	12 February 2019	Incheon	livestock feces (pig)	142	8	5,374,310	50.4	5550	22	94	JAQMUT000000000
MFDS1013836	9 April 2019	Chungcheongnam-do	livestock feces (pig)	275	6	5,685,673	50.5	5819	22	92	JAQMUU000000000
MFDS1013864	31 May 2019	Chungcheongnam-do	livestock feces (pig)	181	15	5,865,149	50.5	6141	22	94	JAQMUV000000000
MFDS1013921	15 November 2019	Jeollabuk-do	animal source foods (beef)	231	11	5,498,083	50.6	5669	22	100	JAQMUW000000000
MFDS1014122	7 January 2019	Gwangju	animal source foods (beef)	177	5	5,462,265	50.3	5487	22	98	JAQMUX000000000
MFDS1014182	9 July 2019	Gyeongsangbuk-do	animal source foods (beef)	235	11	5,557,991	50.5	5767	22	101	JAQMUY000000000
MFDS1015939	10 February 2020	Gwangju	animal source foods (beef)	319	5	5,406,905	50.5	5520	22	96	JAQMUZ000000000
MFDS1016183	24 March 2020	Gyeonggi-do	livestock feces (pig)	438	4	5,064,469	50.9	5087	22	92	JAQMVA000000000
MFDS1016195	27 March 2020	Daejeon	livestock feces (cattle)	246	9	5,789,890	50.5	6119	22	89	JAQMVB000000000
MFDS1016200	27 March 2020	Daejeon	livestock feces (cattle)	524	15	5,393,111	50.9	5542	22	97	JAQMVC000000000
MFDS1016202	5 May 2020	Chungcheongnam-do	livestock feces (pig)	516	10	5,451,302	50.6	5662	22	98	JAQMVD000000000
MFDS1016222	14 April 2020	Chungcheongnam-do	livestock feces (pig)	193	3	5,088,334	50.8	5131	22	92	JAQMVE000000000
MFDS1016224	14 April 2020	Chungcheongnam-do	livestock feces (pig)	198	5	5,304,302	50.8	5334	22	88	JAQMVF000000000
MFDS1016228	12 May 2020	Gyeonggi-do	livestock feces (pig)	387	3	5,064,746	50.9	5087	22	92	JAQMVG000000000
MFDS1016229	12 May 2020	Gyeonggi-do	livestock feces (pig)	197	3	5,068,005	50.9	5081	22	91	JAQMVH000000000
MFDS1016233	5 May 2020	Chungcheongnam-do	livestock feces (pig)	364	6	5,741,335	50.4	5889	22	92	JAQMVI000000000
MFDS1016235	12 May 2020	Gyeonggi-do	livestock feces (pig)	154	8	5,232,588	50.8	5295	22	91	JAQMVJ000000000
MFDS1016416	23 Mar 2020	Jeollabuk-do	animal source foods (pork)	290	7	5,138,577	50.7	5183	22	90	JAQMVK000000000

**Table 2 microorganisms-11-01285-t002:** Serotypes and sequence types of the hybrid STEC/ETEC strains.

Strain Name	Collection Date	Serotype		Sequence Type
MFDS1007784	8 June 2016	O2	H27	ST10
MFDS1009736	31 August 2017	O148	H7	ST1780
MFDS1009760	20 May 2017	O100	H30	ST446
MFDS1009764	4 July 2017	O100	H30	ST446
MFDS1009770	18 July 2017	O100	H30	ST446
MFDS1009773	25 July 2017	O168	H8	ST1021
MFDS1010992	31 January 2018	O168	H8	ST1021
MFDS1012367	28 December 2018	O2	H25	ST670
MFDS1013781	12 February 2019	O100	H30	ST446
MFDS1013782	12 February 2019	O8	H8	ST74
MFDS1013836	9 April 2019	O8	H9	ST1782
MFDS1013864	31 May 2019	O8	H9	ST74
MFDS1013921	15 November 2019	O168	H8	ST1021
MFDS1014122	7 January 2019	O2	H25	ST670
MFDS1014182	9 July 2019	O168	H8	ST1021
MFDS1015939	10 February 2020	O168	H8	ST1021
MFDS1016183	24 March 2020	O155	H21	ST21
MFDS1016195	27 March 2020	O8	H9	ST74
MFDS1016200	27 March 2020	O141	H29	ST785
MFDS1016202	5 May 2020	O155	H21	ST21
MFDS1016222	14 April 2020	O100	H30	ST446
MFDS1016224	14 April 2020	O174	H2	ST726
MFDS1016228	12 May 2020	O155	H21	ST21
MFDS1016229	12 May 2020	O155	H21	ST21
MFDS1016233	5 May 2020	O8	H9	ST74
MFDS1016235	12 May 2020	O100	H30	ST446
MFDS1016416	23 March 2020	O100	H30	ST446

**Table 3 microorganisms-11-01285-t003:** Phage and plasmid replicons of hybrid STEC/ETEC strains.

Strain Name	Phage			Plasmid Replicon
	*stx* Subtype	Completeness	Score	
MFDS1007784	*stx2A*, *stx2B*	intact	150	IncFIB
MFDS1009736	*stx2A*, *stx2B*	intact	150	IncFIB
MFDS1009760	*stx2A*, *stx2B*	intact	150	IncFIl
MFDS1009764	*stx2A*, *stx2B*	intact	150	IncFIl
MFDS1009770	*stx2A*, *stx2B*	intact	150	IncFIl
MFDS1009773	*stx2A*, *stx2B*	intact	110	IncFIB
MFDS1010992	*stx2A*, *stx2B*	intact	150	IncFIB
MFDS1012367	*stx2A*, *stx2B*	questionable	90	IncFIB
MFDS1013781	*stx2A*, *stx2B*	intact	150	IncFIl
MFDS1013782	*stx2A*, *stx2B*	intact	150	IncFIl
MFDS1013836	*stx2A*, *stx2B*	intact	150	IncFIl
MFDS1013864	*stx2A*, *stx2B*	intact	150	IncFIl
MFDS1013921	*stx2A*, *stx2B*	intact	150	IncFIB
MFDS1014122	*stx2A*, *stx2B*	intact	110	IncFIB
MFDS1014182	*stx2A*, *stx2B*	intact	130	IncFIB
MFDS1015939	*stx2A*, *stx2B*	intact	150	IncFIB
MFDS1016183	*stx2A*, *stx2B*	intact	150	IncFIl
MFDS1016195	*stx2A*, *stx2B*	intact	150	IncFIl, IncR, IncX1
MFDS1016200	*stx2A*, *stx2B*	intact	100	IncFIl, IncX1, IncFIB
MFDS1016202	*stx2A*, *stx2B*	intact	150	IncFIl
MFDS1016222	*stx2A*, *stx2B*	intact	140	IncFIl
MFDS1016224	*stx2A*, *stx2B*	intact	150	IncFIl
MFDS1016228	*stx2A*, *stx2B*	intact	150	IncFIl
MFDS1016229	*stx2A*, *stx2B*	intact	150	IncFIl
MFDS1016233	*stx2A*, *stx2B*	intact	150	IncFIl
MFDS1016235	*stx2A*, *stx2B*	intact	140	IncFIl
MFDS1016416	*stx2A*, *stx2B*	intact	150	IncFIl

## Data Availability

Sequence data have been submitted to the publicly accessible NCBI archives (https://ncbi.nlm.nih.gov (accessed on 25 January 2023), including GenBank and Sequence Read Archive (SRA), under the accession numbers listed in Table 1 and Appendix A.

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
