# Peer review of "Genome-Based Characterization of Hybrid Shiga Toxin-Producing and Enterotoxigenic Escherichia coli (STEC/ETEC) Strains Isolated in South Korea, 2016–2020"

_microorganisms, 2023, doi:10.3390/microorganisms11051285_

Round 1

Reviewer 1 Report

The manuscript, entitled “Genomic Characterization and Comparative Analysis of Hybrid Shiga toxin-producing and Enterotoxigenic Escherichia coli (STEC/ETEC) strains isolated in South Korea, 2016-2020” described that the genomes of 27 STEC/ETEC hybrid strains were identified of the virulence and antibiotic resistance genes they and determined their phylogenetic position among other E. coli strains. This study addressed the potential importance of the hybrid E. coli strains for public health. However, due to some drawbacks, my suggestion is major revision.

Major comments:

1. The title of the manuscript is a little of redundant. What’s more, the comparative analysis of the strains was not very clear. It is suggested to make the title short and clearer.

2. Did the authors analyze the cas gene sequence? With the development of CRISPR gene editing technology, the sequence of cas is detected more and more to find more cas sequence. So, it is suggested to add the analysis of cas sequence in these 27 E. coli strains.

3. The sample number of hybrid STEC/ETEC isolates is 27, which is relatively less for a statistical study.

4. The study on the relationship between the genes of virulence factors, antimicrobial profiles and phenotypes of the isolates should also be supplemented.

Minor comments:

1. The labels of strain accession no. in table 1 is confusing. For example, for stain MFDS1007784, there are two accession no. SAMN32786892 and JAQJCX000000000. The meaning of these accession no. should be explained in annotation.

2. In figure 1 and 2, the purpose of them is to show the genes exist in the strains or not. So, 20 to 80 is nonsense in colour key and histogram.

3. In section 3.2, why did the authors use real time PCR to detect genes, instead of normal PCR? If the aim is to identify the virulence and antimicrobial resistance genes, normal PCR can, because some genes are silent in lab culture conditions, and can only express at specific conditions. So, real-time PCR to detect genes is not accuracy. No transcription does not represent no gene sequence in genome, or the subtitle of 3.2 should be revised to make it express clearly.

 Extensive editing of English language required.

Author Response

Response to Reviewer #1 Comments

The manuscript, entitled “Genomic Characterization and Comparative Analysis of Hybrid Shiga toxin-producing and Enterotoxigenic Escherichia coli (STEC/ETEC) strains isolated in South Korea, 2016-2020” described that the genomes of 27 STEC/ETEC hybrid strains were identified of the virulence and antibiotic resistance genes they and determined their phylogenetic position among other E. coli strains. This study addressed the potential importance of the hybrid E. coli strains for public health. However, due to some drawbacks, my suggestion is major revision.

Major comments:

  1. The title of the manuscript is a little of redundant. What’s more, the comparative analysis of the strains was not very clear. It is suggested to make the title short and clearer.

Response: As you recommended, we revised the title as follows:

Title: Genome-Based Characterization of Hybrid Shiga toxin-producing and Enterotoxigenic Escherichia coli (STEC/ETEC) strains isolated in South Korea, 2016-2020

  1. Did the authors analyze the casgene sequence? With the development of CRISPR gene editing technology, the sequence of casis detected more and more to find more cas sequence. So, it is suggested to add the analysis of cas sequence in these 27 E. coli strains.

Response: Thank you for your comments. We additionally analyzed the CRISPRs and cas genes for 27 STEC/ETEC hybrid genomes using CRISPRCasFinder. The contents of this additional analysis were added to the manuscript as follows:

Lines139-141:CRISPRCasFinder (https://crisprcas.i2bc.paris-saclay.fr/CrisprCasFinder/Index) was used to analyze the CRISPRs and cas genes [35].

Lines 190: CRISPR-associated (Cas)

Lines 202-207: The CRISPRFinder server identified a type I CRISPR/Cas system in all hybrid strains. And the most of the STEC/ETEC hybrid strains (26/27) identified the type I-E system. All hybrid strains harbored the cas3 gene, which is the signature of type I CRISPR/Cas systems, responsible for target DNA cleavage and degradation [40]. Furthermore, the most of the STEC/ETEC hybrid strains (26/27) harbored the cas1, cas2, cas5, cas6, and cas7 genes as well as the cas3 gene.

  1. The sample number of hybrid STEC/ETEC isolates is 27, which is relatively less for a statistical study.

Response: Thank you for your comments. This study analyzed all 1,025 pathogenic E. coli strains isolated in South Korea between 2016 and 2020. Only twenty-seven hybrid Shigatoxigenic and Enterotoxigenic Escherichia coli (STEC/ETEC) strains isolated from livestock feces (cattle and pig) and animal source foods (beef, pork, and meat patties) in South Korea. Through genome-based characterization, we have reported for the first time the virulence and antibiotic resistance profile of STEC/ETEC hybrid strains isolated from the livestock feces (cattle and pig) and animal source foods (beef, pork, and meat patties) in South Korea.

  1. The study on the relationship between the genes of virulence factors, antimicrobial profiles and phenotypes of the isolates should also be supplemented.

Response: Thank you for your comments. In this study, we have reported for the first time the virulence and antibiotic resistance profiles of STEC/ETEC hybrid strains isolated from livestock feces (cattle and pigs) and animal source foods (beef, pork, and meat patties) in South Korea. We used the virulence factor database and ResFinder v4.1 to predict virulence factors and antimicrobial resistance genes, respectively, based on genomic sequences. As you recommended, we additionally performed antimicrobial susceptibility tests to determine the phenotypic profile of antimicrobial resistance in STEC/ETEC hybrid strains. The contents of this additional analysis were added to the manuscript as follows:

Lines 218-230: We additionally performed antimicrobial susceptibility tests to determine the phenotypic profile of antimicrobial resistance in STEC/ETEC hybrid strains.  The phenotypic profile of antimicrobial resistance is described in detail in Supplementary Table S3. Comparing the WGS-based AMR genotype to the antimicrobial susceptibility testing-based phenotype for 13 antibiotics (ampicillin, cefepime, cefoxitin, cefotaxime, ceftazidime, chloramphenicol, ciprofloxacin, colistin, nalidixic acid, meropenem, gentamicin, streptomycin, tetracycline) revealed concordant results for 22 of the 27 STEC/ETEC hybrid strains (81.5%). In five STEC/ETEC hybrid strains, only antibiotic resistance genes were identified, but no phenotypes. Although WGS can provide more information about isolates, genomic approaches cannot always predict phenotypes because the level of gene expression and protein production from identified genes may differ between strains. Bacteria have gene silencing mechanisms, and mutations may generate stop codons in the data [41].

Line 379: Table S3: The phenotypic profile of antimicrobial resistance in hybrid STEC/ETEC strains

Minor comments:

  1. The labels of strain accession no. in table 1 is confusing. For example, for stain MFDS1007784, there are two accession no. SAMN32786892 and JAQJCX000000000. The meaning of these accession no. should be explained in annotation.

Response: Sample submission is required as part of data deposit to several NCBI primary data archives including SRA, TSA and WGS. Typically, BioSample data are submitted first and assigned BioSample accession numbers (SAMNxxxxxxxx). To avoid confusion with strain accession No. in Table 1, the BioSample accession numbers are deleted.

  1. In figure 1 and 2, the purpose of them is to show the genes exist in the strains or not. So, 20 to 80 is nonsense in colour key and histogram.

Response: As you recommended, we deleted color key and histogram in Figures 1 and 2.

  1. In section 3.2, why did the authors use real time PCR to detect genes, instead of normal PCR? If the aim is to identify the virulence and antimicrobial resistance genes, normal PCR can, because some genes are silent in lab culture conditions, and can only express at specific conditions. So, real-time PCR to detect genes is not accuracy. No transcription does not represent no gene sequence in genome, or the subtitle of 3.2 should be revised to make it express clearly.

Response: The purpose of this study is to investigate the virulence and antibiotic resistance genes harbored by STEC/ETEC hybrid strains based on the genome sequence. In section 3.2, the real-time PCR for detecting the pathotypes of pathogenic E. coli strains was performed according to the Food Poisoning Cause Investigation Method of the Ministry of Food and Drug Safety (National Institute of Food and Drug Safety Evaluation, Fed. Regist. 11-1471057-000036-19). A multiplex real-time PCR assay can rapidly and simultaneously detect E. coli pathotypes and screen STEC/ETEC hybrid strains. Subsequent WGS analysis of the STEC/ETEC hybrids yielded results that were consistent with the presence of virulence factors (stx2, est). Additionally, detailed results of virulence gene mapping of these hybrid genomes are shown in Supplementary Table S1.

As you recommended, we revised the sentence in lines 191-195 as follows:

Lines 191-195: The initial screening of pathogenic E. coli strains was conducted using real-time PCR. The hybrid STEC/ETEC isolates harbored both Shiga toxin 2 (stx2) and heat-stable enterotoxins (est) encoding genes. Subsequently, we performed virulence gene mapping to identify the various virulence factors present in the STEC/ETEC hybrid genomes (Figure 1).

Reviewer 2 Report

This is a well-written paper - just a few areas where more information/discussion is required. All STEC strains carried only stx2.  Is this normal for Korea?  Especially for cattle-derived strains in North America, stx1 is most common and stx2 relatively rare.  Please discuss.

Table 3 - please add phage status to this table - whether the phage was intact, questionable or incomplete.  Also please ass to the table the proportion of the phage genome present in both the questionable and incomplete categories.  Assuming intact means 100% of phage is present, but if this is incorrect, please describe as well.

L19 This should be 'serogroup' instead of 'serotype'

L142 should be 'virulence genes'

Author Response

Response to Reviewer #2 Comments

This is a well-written paper - just a few areas where more information/discussion is required. All STEC strains carried only stx2.  Is this normal for Korea?  Especially for cattle-derived strains in North America, stx1 is most common and stx2 relatively rare.  Please discuss.

Response: Few studies have reported the virulence profiles of STEC/ETEC hybrid strains isolated from livestock feces (cattle and pigs) and animal source foods (beef, pork, and meat patties) in South Korea. The purpose of this study is to investigate the virulence genes harbored by STEC/ETEC hybrid strains based on the genome sequence. Hybrids of STEC and ETEC strains (STEC/ETEC) have been recently reported in various countries, including Finland, Bangladesh, Sweden, and South Korea, and some have been associated with diarrheal diseases and HUS in humans. Importantly, Shiga toxins 1 and 2 (Stx1 and Stx2, respectively) differ in their virulence and host specificity, with Stx2 being most associated with severe illnesses [hemolytic uremic syndrome (HUS), hospitalization, and bloody diarrhea] in humans. Among hybrid E. coli strains isolated from other countries, it was confirmed that the stx2 gene was dominant in some countries. As you recommended, we added sentence in lines 317-323 as follows:

Lines 317-323: Most of the STEC/ETEC hybrid strains among human and animal isolates in Finland [25] harbored the stx2 gene without stx1 gene. In additionally, STEC/ETEC hybrid strains from diarrheal patients in South Korea [27] and Sweden [28] harbored the stx2 gene. However, the majority of STEC/ETEC hybrid strains in livestock of Bangladesh [26] carried stx1 gene. Our results suggest that the STEC/ETEC hybrid strains isolated in South Korea that contained the stx2 gene may be more dangerous to humans.

Table 3 - please add phage status to this table - whether the phage was intact, questionable or incomplete.  Also please ass to the table the proportion of the phage genome present in both the questionable and incomplete categories.  Assuming intact means 100% of phage is present, but if this is incorrect, please describe as well.

Response: We re-analyzed the bacteriophage sequences using high-quality sequence data obtained from PacBio Sequel (Pacific Bioscience, Menlo Park, CA, USA). PHAge Search Tool Enhanced Release (PHASTER) was used to predict putative prophage regions as “intact (score > 90),” “questionable (score 70-90)”, or “incomplete (score < 70)” based on the proportion of phage-related genes in the identified phage region. Prophages classified as “intact” by PHASTER are the most likely to be complete and functional. Moreover, it has recently been reported that at least some of the “questionable” prophages are still inducible and able to kill the host (de Sousa et al., 2020, doi: 10.1038/s41396-020-0726-z). We revised Table 3 and the sentence in line 148-151, 258-262 as follows:

Lines 148-151: PHASTER was used to predict putative prophage regions as “intact (score > 90)”, “questionable (score 70-90)”, or “incomplete (score < 70)” based on the proportion of phage-related genes in the identified phage region of the assembled hybrid genome.

Lines 258-262: The presence of the majority of stx2 gene sequences was confirmed from phage sequence regions corresponding to “intact” (26/27). And it was confirmed that the sequences corresponding to the stx2 gene in one STEC/ETEC hybrid strain genome (MFDS1012367; score 90) were found in the “questionable” phage region.

L19 This should be 'serogroup' instead of 'serotype'

Response: As you recommended, we revised the sentence in line 19 as follows:

Line 19: The strains represent diverse serogroups

L142 should be 'virulence genes'

Response: As you recommended, we revised the sentence in line 159 as follows:

Line 159: plasmid sequences were annotated to identify virulence genes using the RAST toolkit

Reviewer 3 Report

This manuscript of great interest to healthcare professionals, namely microbiologists, infectious disease specialists and epidemiologists. The team of authors has done a lot of work on the isolation of hybrid STEC/STEC strains from various objects. In my opinion, the studied strains lack one small characteristic, namely, subtypes of shiga toxins are not indicated. The authors can get this information from the WGS data and add it to the table with the characteristics of the strains (Table 3).  I can also recommend the authors to provide data on determining the sensitivity of the studied strains to invitro antimicrobials. It would be a good addition to this work.

Author Response

Response to Reviewer #3 Comments

This manuscript of great interest to healthcare professionals, namely microbiologists, infectious disease specialists and epidemiologists. The team of authors has done a lot of work on the isolation of hybrid STEC/STEC strains from various objects. In my opinion, the studied strains lack one small characteristic, namely, subtypes of shiga toxins are not indicated. The authors can get this information from the WGS data and add it to the table with the characteristics of the strains (Table 3).  I can also recommend the authors to provide data on determining the sensitivity of the studied strains to invitro antimicrobials. It would be a good addition to this work.

Response: The genes encoding Shiga toxin 2 subunit A (stx2A) and Shiga toxin 2 subunit B (stx2B) were placed in the 27 hybrid STEC/STEC strains. As you recommended, we added Shiga toxin 2 subunit in Table3.

Response: As you recommended, we additionally performed antimicrobial susceptibility tests to determine the phenotypic profile of antimicrobial resistance in STEC/ETEC hybrid strains. The contents of this additional analysis and experiment were added to the manuscript as follows:

Lines 94-106: Antimicrobial susceptibility tests were performed using Sensititre KRN6F panels (Trek Diagnostic Systems, Cleveland, OH) following the manufacturer’s instructions. The antimicrobial susceptibility of the isolated strains was determined using the 16 antimicrobials described as follows: amoxicillin–clavulanic acid, ampicillin (AMP), cefoxitin, cefotaxime, ceftazidime, cefepime, chloramphenicol, ciprofloxacin (CIP), colistin, gentamicin, meropenem (MEM), nalidixic acid (NAL), streptomycin, sulfisoxazole, tetracycline (TET), and trimethoprim-sulfamethoxazole. The MIC (Minimum Inhibitory Concentration) value of these antimicrobials was determined with the microbroth dilution method.  The Clinical and Laboratory Standards Institute guidelines and the U.S. National Antimicrobial Resistance Monitoring System were used to interpret susceptibility results expressed as MICs. For these agents, the degree of increase in resistance was determined by referring to the resistance level of the standard strain, ATCC 25922.

Lines 218-230: We additionally performed antimicrobial susceptibility tests to determine the phenotypic profile of antimicrobial resistance in STEC/ETEC hybrid strains. The phenotypic profile of antimicrobial resistance is described in detail in Supplementary Table S3. Comparing the WGS-based AMR genotype to the antimicrobial susceptibility testing-based phenotype for 13 antibiotics (ampicillin, cefepime, cefoxitin, cefotaxime, ceftazidime, chloramphenicol, ciprofloxacin, colistin, nalidixic acid, meropenem, gentamicin, streptomycin, tetracycline) revealed concordant results for 22 of the 27 STEC/ETEC hybrid strains (81.5%). In five STEC/ETEC hybrid strains, only antibiotic resistance genes were identified, but no phenotypes. Although WGS can provide more information about isolates, genomic approaches cannot always predict phenotypes because the level of gene expression and protein production from identified genes may differ between strains. Bacteria have gene silencing mechanisms, and mutations may generate stop codons in the data [41].

Line 379: Table S3: The phenotypic profile of antimicrobial resistance in hybrid STEC/ETEC strains.

Reviewer 4 Report

General Comments:

In general, the manuscript is well-written and informative. The authors provide a clear overview of the global emergence of hybrid diarrheagenic E. coli strains and their potential implications on public health. The study design, including the use of real-time PCR and whole-genome sequencing, is appropriate for investigating the virulence and antibiotic resistance genes of the STEC/ETEC hybrid strains. Additionally, the phylogenetic analysis seems to provide valuable insight into the relatedness of these hybrid strains to other E. coli strains.

However, I would like to suggest a few improvements to further strengthen the article:

1.      In the introduction, it would be beneficial to provide more context on the prevalence and impact of diarrheagenic E. coli infections worldwide, particularly in developing countries. This would help readers understand the significance of the research and the potential implications of the findings.

2.      In the discussion, the authors should mention potential limitations of the study, such as the small number of STEC/ETEC strains analyzed, geographical distribution, and any biases that may be present in the data. Acknowledging these limitations will provide a more balanced perspective on the research.

Other points that need to be addressed before accepting the publication of this work are indicated below:

Major Comments:

Line 169-171: It is unclear whether the presence/absence analysis of virulence genes shown in Figure 1 was performed by real-time PCR or by in silico analysis of the genomes. Please clarify.

It would be more appropriate to include the detection of genes contained in the LAA island in Figure 1 and Supplementary Table S1.

Regarding antimicrobial resistance genes, a histogram figure of the percentage of presence among the 27 strains analyzed would be appropriate. Also, describe the most frequent resistance genes.

Lines 197-201: These results should better indicate the O:H serotype and its frequency.

Lines 237-238: The format of the phylogenetic tree should be improved to allow better analysis.

Importantly, this tree needs to incorporate a Bayesian analysis of population structure to identify lineages. See the following articles as a guide for this analysis:

doi: 10.12688/wellcomeopenres.14694.1

doi: 10.1080/22221751.2019.1595985

Minor Comments:

Line 18: Change "represent" to "belong to"

Line 40: Remove "fatal"

Line 42: Since STEC and ETEC were defined in the previous paragraph, change to "STEC and ETEC are major causes of…"

Line 55: Remove "MGEs and"

Line 64: Change "Consequently" to "Furthermore"

Line 73: Change "molecular" to "genomic"

Lines 74-75: Change "Phylogenetic analysis was performed to assess their phylogeny in a diverse collection of all the major pathogenic E. coli strains" to "Phylogenetic analysis was performed to assess their phylogeny in a collection of E. coli strains from diverse pathotypes."

Line 101: Change "EHEC" to "STEC"

Table 1: Improve the format of the table, either by decreasing the font size or by changing its layout to a horizontal orientation

Lines 274-276: This corresponds to results. Move to lines 178-179.

Lines 306-308: "Importantly, our results emphasize that the hybrid strains of E. coli with STEC and other DEC-associated virulence factors may be more dangerous than STEC alone." This statement needs further discussion with corresponding references.

Author Response

Response to Reviewer #4 Comments

In general, the manuscript is well-written and informative. The authors provide a clear overview of the global emergence of hybrid diarrheagenic E. coli strains and their potential implications on public health. The study design, including the use of real-time PCR and whole-genome sequencing, is appropriate for investigating the virulence and antibiotic resistance genes of the STEC/ETEC hybrid strains. Additionally, the phylogenetic analysis seems to provide valuable insight into the relatedness of these hybrid strains to other E. coli strains.

However, I would like to suggest a few improvements to further strengthen the article:

  1. In the introduction, it would be beneficial to provide more context on the prevalence and impact of diarrheagenic E. coli infections worldwide, particularly in developing countries. This would help readers understand the significance of the research and the potential implications of the findings.

Response: As you recommended, we improved the introduction section and added statements of the prevalence and impact of diarrheagenic E. coli infections worldwide in lines 35 as follows:

Lines 35-38: Diarrheagenic Escherichia coli (DEC) causes 30–40% of acute diarrhea episodes in children <5 years in developing countries [1]. According to the WHO Global Burden of Foodborne Diseases report, >300 million illnesses and nearly 200,000 deaths are caused by DEC globally each year [2].

  1. In the discussion, the authors should mention potential limitations of the study, such as the small number of STEC/ETEC strains analyzed, geographical distribution, and any biases that may be present in the data. Acknowledging these limitations will provide a more balanced perspective on the research.

Response: Thank you for your comments. This study analyzed all 1,025 pathogenic E. coli strains isolated in South Korea between 2016 and 2020. Only twenty-seven hybrid Shigatoxigenic and Enterotoxigenic Escherichia coli (STEC/ETEC) strains isolated from livestock feces (cattle, pig) and animal source foods (beef, pork, meat patties) in South Korea. Through genome-based characterization, we have the first reported the virulence and antibiotic resistance profile of STEC/ETEC hybrid strains isolated from the livestock feces (cattle and pig) and animal source foods (beef, pork, and meat patties) in South Korea. As you recommended, we improved the discussion section and added statements of the potential limitations of the study.

Lines 359-361: In further studies, it may be necessary to investigate the genomic and transcriptome characteristics of STEC/ETEC hybrid strains isolated from diverse ecological and geographical sources in Korea.

Other points that need to be addressed before accepting the publication of this work are indicated below:

Major Comments:

Line 169-171: It is unclear whether the presence/absence analysis of virulence genes shown in Figure 1 was performed by real-time PCR or by in silico analysis of the genomes. Please clarify.

Response: The presence/absence analysis of virulence genes shown in Figure 1 was performed by in silico analysis of the genomes. In section 3.2, the real-time PCR for detecting the pathotypes of pathogenic E. coli strains was performed according to the Food Poisoning Cause Investigation Method of the Ministry of Food and Drug Safety (National Institute of Food and Drug Safety Evaluation, Fed. Regist. 11-1471057-000036-19). A multiplex real-time PCR assay can rapidly and simultaneously detect E. coli pathotypes and screen STEC/ETEC hybrid strains. Subsequent WGS analysis of the STEC/ETEC hybrids yielded results that were consistent with the presence of virulence factors (stx2, est). Additionally, detailed results of virulence gene mapping of these hybrid genomes are shown in Supplementary Table S1.

As you recommended, we revised the sentence in lines 191-195 as follows:

Lines 191-195: The initial screening of pathogenic E. coli strains was conducted using real-time PCR. The hybrid STEC/ETEC isolates harbored both Shiga toxin 2(stx2) and heat-stable enterotoxins (est) encoding genes. Subsequently, we performed virulence gene mapping to identify the various virulence factors present in the STEC/ETEC hybrid genomes (Figure 1).

It would be more appropriate to include the detection of genes contained in the LAA island in Figure 1 and Supplementary Table S1.

As you recommended, we added the contents of Figure S1 to Figure 1, as well as LAA-related genes to supplementary Table S1.

Regarding antimicrobial resistance genes, a histogram figure of the percentage of presence among the 27 strains analyzed would be appropriate. Also, describe the most frequent resistance genes.

Response: As you recommended, we added the sentence in lines 214-218 as follows:

Lines 214-218: High rates of resistance gene were observed for tetracycline (55.6%), doxycycline (55.6%), chloramphenicol (51.9%), sulfamethoxazole (51.9%), florfenicol (48.1%), streptomycin (48.1%), amoxicillin (44.4%), ampicillin (44.4%), and piperacillin (44.4%). Especially, the tetA and tetB genes found at the highest frequency in hybrid E. coli strains.

Lines 197-201: These results should better indicate the O:H serotype and its frequency.

Response: As you recommended, we revised the sentence in lines 246-250 as follows:

Lines 246-250: Based on in silico serotyping, the 27 hybrid E. coli strains belonged to eight distinct O:H serogroups [O100:H30 (n = 7), O8:H9 (n = 5), O168:H8 (n = 5), O155:H21 (n = 4), O2:H25 (n = 3), O141:H29 (n = 1), O148:H7 (n = 1), and O174:H2 (n = 1)]. Especially, O100:H30 serogroup was found at the highest frequency in hybrid E. coli strains.

Lines 237-238: The format of the phylogenetic tree should be improved to allow better analysis.

Importantly, this tree needs to incorporate a Bayesian analysis of population structure to identify lineages. See the following articles as a guide for this analysis:

doi: 10.12688/wellcomeopenres.14694.1

doi: 10.1080/22221751.2019.1595985

Response: Thank you for your comments. We additionally performed the population structure analysis for 27 STEC/ETEC hybrid genomes using RhierBAPs. The contents of this additional analysis were revised and added to the manuscript as follows:

Line 161: Phylogenetic analysis and Population structure analysis

Lines 171-172: The population structure analysis was performed using RhierBAPs [39].

Line 275: Phylogenetic analysis and Population structure analysis

Lines 278-290: The 187 genome datasets included the sequencing results of 160 pathogenic E. coli as well as 27 hybrid STEC/ETEC genomes, which were deposited in the NCBI database. Phylogenetic tree analysis revealed that these hybrids were closely related to certain ETEC (21 strains, 77.8%) and STEC (six strains, 22.2%) strains, implying the potential acquisition of Stx-phages and/or ETEC virulence genes during their emergence (Figure 3A). In addition, the population structure of the 187 genome datasets was defined using the RhierBAPSs, which divided the genome datasets into 6 primary sequence clusters (Bayesian analysis of population structure [BAPS] hierarchical level 1). These were further subdivided into 28 lineages (BAPS level 2) (Figure 3B). The results showed that of the total, 21 hybrid strains closely related to ETEC were divided into six groups (three level 1, five level 2), while the remaining six hybrids correlated with STEC were divided into two groups (two level 1, two level 2).

Lines 293-300: Figure 3. Phylogenetic analysis and population structure analysis of hybrid STEC/ETEC strains. (A) Strains are colored based on the pathogenic E. coli groups: blue, STEC strains; pink, ETEC strains; gray, EPEC strains; green, EAEC strain; and brown, EIEC strains. The pink and blue bars represent 21 and six STEC/ETEC hybrid strains that are closely related to specific ETEC and STEC strains, respectively. (B) Sequence clusters (1 to 6) are indicated in the outer colored dot, which are further divided into 28 lineages (inner ring). Of the total, 21 hybrid strains closely related to ETEC were divided into six groups (three level 1, five level 2), while the remaining six hybrids correlated with STEC were divided into two groups (two level 1, two level 2).

Minor Comments:

Line 18: Change "represent" to "belong to"

Response: As you recommended, we revised the sentence in line 19 as follows:

Line 19: The strains belong to diverse

Line 40: Remove "fatal"

Response: As you recommended, we revised the sentence in line 43 as follows:

Line 43: foodborne pathogens that raise public health concerns and cause several outbreaks

Line 42: Since STEC and ETEC were defined in the previous paragraph, change to "STEC and ETEC are major causes of…"

Response: As you recommended, we revised the sentence in line 45 as follows:

Line 45: STEC and ETEC are major causes of…

Line 55: Remove "MGEs and"

Response: As you recommended, we revised the sentence in line 57 as follows:

Line 57: the transmission of virulence genes via horizontal gene transfer

Line 64: Change "Consequently" to "Furthermore"

Response: As you recommended, we revised the sentence in line 66 as follows:

Line 66: Furthermore, hybrids of STEC and ETEC strains

Line 73: Change "molecular" to "genomic"

Response: As you recommended, we revised the sentence in line 75 as follows:

Line 75: The genomic properties of these strains were investigated

Lines 74-75: Change "Phylogenetic analysis was performed to assess their phylogeny in a diverse collection of all the major pathogenic E. coli strains" to "Phylogenetic analysis was performed to assess their phylogeny in a collection of E. coli strains from diverse pathotypes."

Response: As you recommended, we revised the sentence in lines 76-77 as follows:

Lines 76-77: Phylogenetic analysis was performed to assess their phylogeny in a collection of E. coli strains from diverse pathotypes

Line 101: Change "EHEC" to "STEC"

Response: As you recommended, we revised the sentence in line 115 as follows:

Line 115: VT1 and VT2 (STEC)

Table 1: Improve the format of the table, either by decreasing the font size or by changing its layout to a horizontal orientation

Response: As you recommended, we improved the format of table by decreasing the font size and by changing its layout to a horizontal orientation

Lines 274-276: This corresponds to results. Move to lines 178-179.

Response: As you recommended, we moved the sentence as follows:

Lines 200-202: Importantly, we also detected genes encoding LAA, such as hes, iha, lesP, and agn43, which are related to STEC pathogenicity (Supplementary Figure S1). The most prevalent gene was iha (59.3%), followed by agn43, hes, and lesP that were present in 25.9, 18.5, and 11.1% of the hybrid strains, respectively.

Lines 306-308: "Importantly, our results emphasize that the hybrid strains of E. coli with STEC and other DEC-associated virulence factors may be more dangerous than STEC alone." This statement needs further discussion with corresponding references.

Response: As you recommended, we improved the conclusion section and added sentence in lines 368-373 as follows:

Lines 368-373: In addition, we identified adhesion and colonization factors in the human intestine required for STEC pathogenesis. Most DEC is subdivided into several pathotypes based on the presence of specific virulence traits directly related to disease development [3, 4]. Importantly, our results emphasize that the hybrid strains of E. coli with STEC and other DEC-associated virulence factors may be more dangerous than STEC alone [14, 20-28].

Round 2

Reviewer 1 Report

The authors of the manuscript “Genome-based Characterization of Hybrid Shiga toxin-producing and Enterotoxigenic Escherichia coli (STEC/ETEC) strains isolated in South Korea, 2016-2020” have revised the text carefully.

1. The names of genus and species should be in italics. For example, in line 35 “Escherichia coli” should be expressed as “Escherichia coli”.

2. In section of 2.2 Antimicrobial susceptibility tests, the abbreviations of some antimicrobials were provided, while some were not. It is suggested to keep them consistent.

3. The section of 2.3 is the methods of RT-qPCR, but the title is “Polymerase Chain Reaction (PCR)-based Identification of Hybrid Strains”. The title is inappropriate. In addition, the primers used should also be attached.

4. The resolution of figure 1 is low. Please provide vector diagram with high resolution. Also, for figure 2.

5. The format of references should be consistent. For example, the years are in bold in some references, but some are not. Please check them carefully and revise the errors.

N/A

Author Response

  1. The names of genus and species should be in italics. For example, in line 35 “Escherichia coli” should be expressed as “Escherichia coli”.

Response: As you recommended, we revised the sentence in line 35 as follows:

Line 35: Escherichia coli

  1. In section of 2.2 Antimicrobial susceptibility tests, the abbreviations of some antimicrobials were provided, while some were not. It is suggested to keep them consistent.

Response: As you recommended, we revised the sentence in line 98-101 as follows:

Lines 98-101: amoxicillin–clavulanic acid, ampicillin, cefoxitin, cefotaxime, ceftazidime, cefepime, chloramphenicol, ciprofloxacin, colistin, gentamicin, meropenem, nalidixic acid, strep-tomycin, sulfisoxazole, tetracycline, and trimethoprim-sulfamethoxazole.

  1. The section of 2.3 is the methods of RT-qPCR, but the title is “Polymerase Chain Reaction (PCR)-based Identification of Hybrid Strains”. The title is inappropriate. In addition, the primers used should also be attached.

Response: Thank you for your comments. In section 2.3, the real-time PCR was performed using a PowerCheckTM 20/15 Pathogen Multiplex Real-time PCR kit (Kogene Biotech Co., Ltd., Seoul, Korea) to detect virulence genes. The real-time PCR for detecting the pathotypes of pathogenic E. coli strains was performed according to the Food Poisoning Cause Investigation Method of the Ministry of Food and Drug Safety (National Institute of Food and Drug Safety Evaluation, Fed. Regist. 11-1471057-000036-19). However, since the detection of virulence genes uses a commercial kit and the primer sequence is not disclosed, we have included the manufacturer and kit information in lines 111-116.

As you recommended, we revised the 2.3 section title in line 107 as follows:

Line 107: Real-time PCR based Identification of Hybrid Strains.

  1. The resolution of figure 1 is low. Please provide vector diagram with high resolution. Also, for figure 2.

Response: As you recommended, we improved the resolution of figure 1 and figure 2.

  1. The format of references should be consistent. For example, the years are in bold in some references, but some are not. Please check them carefully and revise the errors.

Response: As you recommended, we checked and revised the format of references

Reviewer 4 Report

The authors have incorporated all the comments and suggestions made by this reviewer, resulting in a much more structured and clearer version of the manuscript.

However, there are a few minor comments to address.

Firstly, on line 35, "Escherichia coli" should be in italics.

Secondly, Supplementary Table 1 has overlapping columns on the LAA genes that were added and needs to be corrected.

Thirdly, on line 284, "RhierBAPSs" should be changed to "RhierBAPS".

Finally, for Figure 3, I suggest that the authors indicate the BAPS levels on the same tree with branches colored according to their BAPS clustering, or alternatively, use iTOL to include data and points outside the tree using symbols such as squares or triangles. Ultimately, it is up to the authors to decide whether or not to make this change to improve the format of the figure."

Author Response

The authors have incorporated all the comments and suggestions made by this reviewer, resulting in a much more structured and clearer version of the manuscript.

However, there are a few minor comments to address.

Firstly, on line 35, "Escherichia coli" should be in italics.

Response: As you recommended, we revised the sentence in line 35 as follows:

Line 35: Escherichia coli

Secondly, Supplementary Table 1 has overlapping columns on the LAA genes that were added and needs to be corrected.

Response: Thank you for your comments. We revised the Supplementary Table 1

Thirdly, on line 284, "RhierBAPSs" should be changed to "RhierBAPS".

Response: As you recommended, we revised the sentence in line 284 as follows:

Line 284: RhierBAPS

Finally, for Figure 3, I suggest that the authors indicate the BAPS levels on the same tree with branches colored according to their BAPS clustering, or alternatively, use iTOL to include data and points outside the tree using symbols such as squares or triangles. Ultimately, it is up to the authors to decide whether or not to make this change to improve the format of the figure."

Response: Thank you for your comments. We indicated the BAPS levels on the same tree using circular colored (six level 1) and triangles colored (twenty-eight level 2) according to their BAPS clustering.